# MLP-MFF: Lightweight Pyramid Fusion MLP for Ultra-Efficient End-to-End Multi-Focus Image Fusion

**DOI:** 10.3390/s25165146

**Published:** 2025-08-19

**Authors:** Yuze Song, Xinzhe Xie, Buyu Guo, Xiaofei Xiong, Peiliang Li

**Affiliations:** 1National Ocean Technology Center, Tianjin 300112, China; notcsongyuze@163.com; 2Key Laboratory of Ocean Observation Technology, Ministry of National Resources, Tianjin 300112, China; 3State Key Laboratory of Ocean Sensing, Ocean College, Zhejiang University, Zhoushan 316021, China; xiexinzhe@zju.edu.cn (X.X.); lipeiliang@zju.edu.cn (P.L.); 4Donghai Laboratory, Zhoushan 316021, China; 5Hainan Institute, Zhejiang University, Sanya 572025, China; 6South China Sea Ecological Center, Ministry of Natural Resources, Guangzhou 510275, China; xiongxiaofei124@126.com; 7Hainan Observation and Research Station of Ecological Environment and Fishery Resource in Yazhou Bay, Sanya 572024, China

**Keywords:** deep learning, multi-focus image fusion, lightweight network, end to end, Multi-Layer Perceptron

## Abstract

Limited depth of field in modern optical imaging systems often results in partially focused images. Multi-focus image fusion (MFF) addresses this by synthesizing an all-in-focus image from multiple source images captured at different focal planes. While deep learning-based MFF methods have shown promising results, existing approaches face significant challenges. Convolutional Neural Networks (CNNs) often struggle to capture long-range dependencies effectively, while Transformer and Mamba-based architectures, despite their strengths, suffer from high computational costs and rigid input size constraints, frequently necessitating patch-wise fusion during inference—a compromise that undermines the realization of a true global receptive field. To overcome these limitations, we propose MLP-MFF, a novel lightweight, end-to-end MFF network built upon the Pyramid Fusion Multi-Layer Perceptron (PFMLP) architecture. MLP-MFF is specifically designed to handle flexible input scales, efficiently learn multi-scale feature representations, and capture critical long-range dependencies. Furthermore, we introduce a Dual-Path Adaptive Multi-scale Feature-Fusion Module based on Hybrid Attention (DAMFFM-HA), which adaptively integrates hybrid attention mechanisms and allocates weights to optimally fuse multi-scale features, thereby significantly enhancing fusion performance. Extensive experiments on public multi-focus image datasets demonstrate that our proposed MLP-MFF achieves competitive, and often superior, fusion quality compared to current state-of-the-art MFF methods, all while maintaining a lightweight and efficient architecture.

## 1. Introduction

Modern optical imaging systems inherently face the challenge of limited depth of field (DoF) when capturing 3D scenes. This means that in a single exposure, only objects within a specific focal plane appear sharp, while areas outside this plane become progressively blurred with increasing defocus. This physical limitation severely constrains the completeness and visual quality of image information, particularly impacting applications with stringent requirements for all-in-focus clarity, such as microscopic imaging [1,2], machine vision [3] and industrial inspection [4]. To overcome the drawback of insufficient DoF in single images, multi-focus image fusion (MFF) technology has emerged. This technique aims to combine a series of partially focused images of the same scene, acquired at different focal settings, into a single image that maintains good focus across the entire spatial range, thereby significantly expanding the information content and usability of the image [5].

To achieve all-in-focus image construction, researchers have proposed various MFF algorithms. These algorithms can be broadly categorized into three types based on their processing principles and image domains: spatial domain-based fusion methods, transform domain-based fusion methods and the more recently developed deep learning-based fusion methods.

Spatial domain methods [6,7,8,9,10] directly perform focus analysis and information selection at the pixel or local region level. These methods typically rely on hand-crafted focus measure operators and directly select or weighted-average pixels from source images based on the evaluation results. Their advantages include simple implementation and lower computational cost. However, they often produce artifacts at focus boundaries and are sensitive to noise and slight registration errors in the source images.

In contrast, transform domain methods [11,12,13,14,15,16,17,18] convert images into specific coefficient domains (e.g., wavelet or Laplacian pyramid domains) for feature extraction and fusion, then inverse transform them back to the spatial domain. This approach better separates high-frequency and low-frequency information, helping to preserve details like textures and edges in the fused image. Despite these advantages, transform domain methods generally involve higher computational complexity and significantly longer fusion times.

With the rapid advancements in deep learning, its data-driven powerful feature-learning capabilities have offered new solutions to the challenges faced by traditional fusion methods. Deep learning-based MFF methods can generally be summarized into two main paradigms. The first category consists of decision map-based deep learning methods. These methods draw inspiration from traditional spatial domain fusion approaches, utilizing deep neural networks to automatically learn and generate a focus decision map, indicating from which source image each pixel or region should be extracted. This approach often yields more interpretable results and can effectively leverage insights from existing traditional methods, for example, by integrating morphological processing or conditional random fields in post-processing to optimize the decision map.

The other paradigm is end-to-end deep learning regression methods. Unlike decision map-based methods, these directly take a series of defocused source images as input and, through an end-to-end trained deep neural network, directly regress to the final all-in-focus fused image. By constructing complex nonlinear mapping relationships, this method automatically learns the entire fusion process from input to output, without explicit focus measure calculations or fusion rule design. This holds the promise of generating higher-quality, more natural fusion results and simplifying the overall fusion workflow. The end-to-end regression paradigm, with its powerful adaptive learning and feature representation capabilities, has demonstrated significant advantages in suppressing artifacts, preserving details, enhancing the visual quality of fused images, and handling complex scenes, making it a current research hot topic and development trend in the MFF field.

However, early deep learning-based end-to-end fusion methods largely relied on Convolutional Neural Networks (CNNs) [19,20]. While CNNs excel at extracting local features, their inherent limitations in capturing long-range dependencies and global contextual information have become increasingly apparent in complex fusion tasks. This has sparked interest in alternative deep learning architectures. Recently, models such as Transformers [21,22], known for their superior long-range interaction modeling capabilities, and Diffusion models [23], which offer powerful generative capabilities for high-quality image synthesis, are emerging as promising avenues for further advancing MFF. Although these new paradigms aim to overcome the limitations of CNNs and pave the way for more robust and visually compelling fusion results, they also face their own challenges. While Transformer models excel at capturing global dependencies, their core self-attention mechanism leads to substantial computational and memory overhead, especially when processing high-resolution images, which limits their practical application. Furthermore, their efficiency in capturing local image details and textures may be lower compared to CNNs. Diffusion models exhibit extraordinary performance in image generation, but their inference speed is extremely slow, requiring hundreds or even thousands of iterative steps to generate an image, which makes it difficult to meet real-time or near real-time fusion demands. Simultaneously, their training and inference processes consume significant computational resources, and as generative models, their goal is to create content rather than precisely select and integrate information from source images, which may lead to inconsistencies in the fusion process. The recently emerging Mamba model, as a novel architecture based on State Space Models (SSM), has shown great potential in sequence modeling. Mamba, with its linear computational complexity and long-range dependency modeling capability, is expected to overcome the computational efficiency bottlenecks of Transformer models when processing long sequence data, offering a more efficient solution. However, due to architectural limitations, some of the aforementioned models typically only support fixed input image sizes [24,25]. When higher-resolution images need to be fused during the inference phase, they can only be processed by dividing them into patches and then merging them. This approach cannot truly achieve full-image long-range interaction, and such a compromise contradicts the initial purpose for which these architectures were introduced into this field.

Recently, models composed solely of Multi-Layer Perceptrons (MLPs) have shown new potential in the vision domain [26,27], also becoming an emerging trend, particularly with the proposal of some MLP architectures that support flexible input scales, which are expected to solve the aforementioned problems. This paper aims to delve into the application potential of this neural network architecture in the task of MFF. Our research seeks to demonstrate that a carefully designed lightweight MLP-based network can effectively address the challenges faced by MFF (Figure 1), thereby offering a novel perspective and a more competitive alternative to existing methods. The main contributions of this paper are threefold:We propose a lightweight, end-to-end MFF network based on Pyramid Fusion MLP, which can handle flexible input scales, learn multi-scale feature representations, and capture long-range dependencies.We propose a Dual-Path Adaptive Multi-scale Feature-Fusion Module (DAMFFM-HA) that integrates hybrid attention mechanisms and adaptive weight allocation to effectively fuse multi-scale features, thereby enhancing fusion performance.Extensive experiments on publicly available datasets demonstrate the effectiveness and superiority of our proposed method compared to state-of-the-art MFF methods.

The remainder of this paper is organized as follows: Section 2 provides a comprehensive review of related works in MFF, encompassing traditional methods, deep learning-based approaches, existing MLP-based network architecture, and existing MLP-based image-fusion networks. Section 3 presents our proposed method. Section 4 delivers comprehensive experimental results and comparisons with state-of-the-art methods. Section 5 discusses the limitations and potential future directions of our work. Finally, Section 6 concludes the paper.

## 2. Related Works

### 2.1. Conventional Image-Fusion Methods

**Transform-domain methods.** Transform-domain methods include methods utilizing pyramids [11,12], wavelets [13,14], Discrete Cosine Transform (DCT) [15], Non-Subsampled Contourlet Transform (NSCT) [16,29], and Sparse Representation (SR) [17,18]. While these methods offer the distinct advantage of integrating feature information across diverse frequency spectra, they inherently demand the judicious manual selection of appropriate transformations and fusion rules.

**Spatial-domain methods.** Block-based approaches [6,7] divide images into patches and select blocks with the highest activity measures, providing computational efficiency but potentially introducing blocking artifacts at patch boundaries. Region-based methods [8] perform segmentation first and then apply fusion rules to each region, maintaining spatial coherence but being vulnerable to segmentation errors. Advanced pixel-based techniques using SIFT features or defocus estimation [9,10] provide more sophisticated selection criteria but are computationally intensive and noise-sensitive.

### 2.2. Deep Learning-Based Image-Fusion Methods

**Decision map-based methods.** This direction was pioneered by CNN [19], which utilized convolutional networks for this purpose. Subsequent research has advanced performance through various strategies: MSFIN [20] and GEU-Net [30] leveraged multi-scale features; SESF [31] achieved unsupervised fusion by computing pixel-level spatial frequencies within feature maps; SMFuse [32] introduced a self-supervised learning-based fusion technique; MFIF-GAN [33] significantly improved decision map quality by employing adversarial learning and α-matte modeling. More recent studies, such as [34], have attempted to integrate Transformers for global image modeling to overcome the inherent limitations of Convolutional Neural Networks. ZMFF [35] realized zero-shot fusion by utilizing Deep Image Prior. Furthermore, BridgeMFF [24] proposed a dual-adversarial-based decision map-refinement method. In a notable recent work, Ref. [25] introduced the Wavelet Mamba Module to the MFF field, combining it with deep priors to further enhance the accuracy of decision maps. MFIF-STCU-Net [36] achieved state-of-the-art results by leveraging a hybrid U-Net and Transformer architecture to synthesize training data using a depth estimation model. Building on this, DMANet [37] introduced explicit defocus blur modeling into the MFF process, enhancing both interpretability and performance. Most recently, LSKN-MFIF [38] surpassed existing methods with a dynamically adjusting “large selective kernel” module that captures both global and local features more accurately.

**End-to-end methods.** IFCNN [39] and U2Fusion [40] were among the first to apply end-to-end methods for MFF. MFF-GAN [41] proposed an unsupervised generative adversarial network with adaptive and gradient joint constraints for MFF. More recently, SwinFusion [22], SwinMFF [21] and FusionDiff [23] have applied Transformers and diffusion models to this domain. Recently, DDBFusion [42] further proposed a unified image decomposition and fusion framework based on dual decomposition and Bézier curves, enhancing the filtering capability for redundant information. StackMFF [5] extended the approach to image stacks using 3D CNNs and synthesized training data. Meanwhile, MMAE [43] introduced a mask attention mechanism to filter out redundant information. A different direction was taken by LFDT-Fusion [44], which proposed an efficient latent feature-guided diffusion model that works in a compressed latent space.

The proposed MLP-MFF adopts an end-to-end approach rather than a decision map-based one. Beyond the aforementioned advantages, this choice streamlines the processing pipeline, significantly reducing error accumulation that might arise from intermediate steps. This simplification contributes to a more user-friendly and effective solution in practical applications, demonstrating greater potential for deployment in production environments.

### 2.3. MLP-like Architecture in Computer Vision

The MLP-Mixer [45] introduced a novel vision architecture based on Multi-Layer Perceptrons (MLPs), demonstrating performance competitive with CNNs and Transformers in computer vision tasks. This seminal work established that convolutions and attention mechanisms are not indispensable for achieving strong performance. Following this, several variants like ResMLP [46], RepMLPNet [47], EAMLP [48], ViP [49], sMLP-Net [50], and Strip-MLP [51] were proposed. These aim for a better balance between performance and efficiency, yet similar to many Transformer-based networks, they are restricted to fixed-dimension inputs, precluding their direct application in downstream dense image prediction tasks.

Researchers have attempted to address this limitation by substituting spatial MLPs with alternative spatial aggregation operations, such as those found in AS-MLP [52], S2-MLP [53], Shift MLP [54], and CycleMLP [27]. However, these approaches sacrifice global receptive fields and struggle to capture multi-scale information. Furthermore, hybrid architectures combining convolutions and MLPs, including Wave-MLP [55], ATMNet [56], and RaMLP [57], have been proposed to mitigate the aforementioned issues, albeit at the cost of increased network complexity.

In contrast, PFMLP [26] offers a more competitive and efficient pure-MLP architecture, termed Pyramid Fusion MLP, to overcome these limitations. Specifically, each block within PFMLP incorporates multi-scale pooling and fully connected layers to generate a feature pyramid, which is then fused via upsampling layers and additional fully connected layers. Employing diverse downsampling rates enables the acquisition of varying receptive fields, allowing the model to concurrently capture long-range dependencies and fine-grained details. This fully leverages the potential of global contextual information, thereby enhancing the model’s spatial representation capabilities. To our knowledge, PFMLP is currently one of the few MLP architectures that simultaneously supports variable input dimensions, maintains a global receptive field, facilitates multi-scale information interaction, and remains purely MLP-based. The backbone network presented in this paper is implemented based on PFMLP.

### 2.4. MLP-Based Network in Image Fusion

While CNNs and Transformer-based networks have been extensively explored in image fusion, MLP-based networks remain relatively underexplored in this domain. Nevertheless, recent studies have begun to introduce MLP-based architectures into image fusion.

For instance, FusionMLP [58], building upon MAXIM [59], proposes the first MLP-based multi-scene image-fusion network. Similarly, CMFuse [60] introduces a novel infrared and visible image-fusion network that leverages a hybrid CNN and MLP architecture to effectively model long-range dependencies and facilitate cross-modal information exchange.

In contrast to these prior works, our research specifically designs an MLP-based network for MFF. Furthermore, our primary focus is on achieving an optimal balance between model efficiency and fusion performance.

## 3. Methods

### 3.1. Overall Architecture

As the backbone of the proposed MLP-MFF, PFMLP is a multi-stage architecture capable of capturing multi-scale context and a global receptive field, all without relying on convolution layers. This allows it to effectively handle variable-sized inputs. As illustrated in Figure 2, the framework first takes two input images of size 3×3 and uses 3×3 convolutions to map them into feature maps of size 8×H/2×W/2.

The network then proceeds through four stages to capture both local and global spatial information and facilitate inter-channel communication. Each stage begins with a single PFMLP block that incorporates downsampling, reducing the image height and width by half while doubling the channel count. The PFMLP block is utilized to enable information exchange across different spatial locations and channels. As depicted in Figure 3, each PFMLP block consists of a Pyramid Fusion (PF) module and a Single Scale (SS) module, with residual connections applied after each module.

Following each stage, the outputs from both images are fed into the proposed Dual-Path Adaptive Multi-scale Feature-Fusion Module based on Hybrid Attention (DAMFFM-HA) for multi-scale fusion. Finally, the fused image is progressively restored to its original dimensions through four decoders, each comprising bilinear interpolation and convolutional layers.

The proposed MLP-MFF is a MFF network characterized by its global and multi-scale receptive fields, offering flexible handling of various image sizes. Unlike SwinFusion [22] and SwinMFF [21], MLP-MFF does not necessitate patch-wise fusion, thereby genuinely leveraging a global receptive field for MFF.

### 3.2. Pyramid Fusion MLP Block

The Pyramid Fusion MLP Block comprises a Pyramid Fusion (PF) Module and a Single Scale (SS) Module, as depicted in Figure 3.

**Pyramid Fusion Module (PF Module).** The PF module generates a feature pyramid using four branches, each employing a pooling layer with a distinct kernel size. Unlike conventional pooling layers, the PF module utilizes learnable pooling layers to more effectively aggregate local visual cues. Following the acquisition of the feature pyramid, a channel fully connected (FC) layer maps the input *C* channels to $C^{\prime }$ channels. Subsequently, nearest neighbor sampling is applied to obtain multiple feature maps, which are then summed and passed through a GELU activation function. Finally, another channel FC layer transforms the summation result into the final feature representation.

**Single Scale Module (SS Module).** The SS module, a variant of the channel MLP, exists in two forms: a naive version and a down-sample version. The naive SS module consists of two-channel MLPs with a GELU activation between them, and a pooling layer (without down-sampling) for information aggregation. The down-sample SS module introduces an additional branch comprising a depth-wise convolution layer and a channel FC layer. Furthermore, the pooling layer in the original branch is replaced with a depth-wise convolution layer. The final output is obtained by summing the results of both branches, thereby minimizing spatial information loss during down-sampling and facilitating more efficient feature extraction.

For a comprehensive description of the Pyramid Fusion MLP Block, readers are referred to PFMLP [26].

### 3.3. Dual-Path Adaptive Multi-Scale Feature-Fusion Module

To effectively fuse multi-scale features from two multi-focus images, this paper proposes a Dual-Path Adaptive Multi-scale Feature-Fusion Module (DAMFFM-HA), as shown in Figure 4. This module integrates hybrid attention mechanisms and adaptive weight allocation strategies to capture and fuse complementary information between different images, thereby improving fusion quality. The DAMFFM-HA comprises two core components: an attention module responsible for importance assessment and enhancement of input features, and a fusion block that executes adaptive feature-fusion operations.

**Hybrid Attention Module.** The attention module employs a hybrid attention mechanism for two-dimensional feature processing, which consists of channel attention for capturing global feature dependencies and spatial attention for emphasizing important spatial locations within feature maps.

The channel attention mechanism is designed to model interdependencies among feature channels and adaptively recalibrate channel-wise feature responses by explicitly modeling channel relationships. For input feature maps $F\in R^{C\times H\times W}$, global average pooling is applied to obtain $F_{gap}\in R^{C\times 1\times 1}$, which aggregates spatial information across each channel. This is followed by dimension reduction convolution with reduction ratio of 8, ReLU activation, dimension expansion convolution, and Sigmoid activation to generate channel attention weights $A_{c}\in R^{C\times 1\times 1}$.

The spatial attention mechanism focuses on identifying and highlighting informative spatial regions within feature maps to suppress irrelevant background information and enhance focus-relevant areas. The spatial attention process applies 1×1 convolution for input feature dimension reduction, followed by batch normalization and ReLU activation, then another 1×1 convolution to generate single-channel spatial attention maps $A_{s}\in R^{1\times H\times W}$, which are finally normalized through Sigmoid function to produce spatial importance weights.

The final attention-enhanced feature representation is obtained by sequentially applying both attention mechanisms: $F_{att}=F⊙A_{c}⊙A_{s}$, where ⊙ denotes element-wise multiplication.

**Adaptive Fusion Block.** The adaptive fusion block serves as the core component of DAMFFM-HA, responsible for adaptively fusing dual-path features enhanced by attention. The module processes input features $F_{1}$ and $F_{2}$ through independent attention modules to obtain enhanced features $F_{att1}$ and $F_{att2}$, ensuring each feature path is optimized according to its inherent characteristics.

The module generates adaptive weights by concatenating two enhanced features along the channel dimension to form $F_{concat}=\left[F_{att1},F_{att2}\right]\in R^{2C\times H\times W}$, then employs a weight generation network consisting of two 3×3 convolutions, batch normalization, GELU activation, and final Softmax normalization to produce pixel-level fusion weights $W=\left[W_{1},W_{2}\right]\in R^{2\times H\times W}$ where $W_{1}+W_{2}=1$.

The weighted fusion and feature-refinement process first performs adaptive weighted fusion $F_{weighted}=W_{1}⊙F_{att1}+W_{2}⊙F_{att2}$, while simultaneously processing concatenated features through a fusion convolutional network to generate refined features $F_{refined}=\phi \left(F_{concat}\right)$, where ϕ(·) represents a feature-refinement network composed of two 3×3 convolutions, batch normalization, and GELU activation functions, with final fused features obtained through residual connection as $F_{fused}=F_{weighted}+F_{refined}$.

### 3.4. Decoder

The decoder is responsible for progressively reconstructing the fused image from multi-scale fused features to the original resolution, employing a five-stage upsampling architecture.

**Upsampling Block Design.** Each decoder stage employs an upsampling block that combines bilinear interpolation with convolutional operations. This module first doubles the spatial dimensions through bilinear interpolation, followed by two consecutive 3×3 convolutions with batch normalization and GELU activation functions, ensuring smooth feature transitions and reducing aliasing effects during upsampling.

For an input feature map $F_{in}\in R^{C_{in}\times H\times W}$, the upsampling block operates as:$$F_{up}={\mathrm{Conv}}_{3\times 3}\left(\mathrm{BN}\left(\mathrm{GELU}\left({\mathrm{Conv}}_{3\times 3}\left(\mathrm{BN}\left(\mathrm{Interpolate}\left(F_{in},\mathrm{scale}=2\right)\right)\right)\right)\right)\right)$$
where Interpolate(·) represents bilinear interpolation with a scale factor of 2, and the output feature map has dimensions $C_{out}\times 2H\times 2W$.

**Progressive Reconstruction with Skip Connections.** The decoder incorporates skip connections between corresponding encoder and decoder stages to preserve fine-grained spatial details that may be lost during encoding. The decoder processes multi-scale fused features in the following sequence:

Stage 4 to 3: The deepest fused features $X^{4}$ are upsampled through Decoder4 and element-wise added with $X^{3}$: $F_{up4}=\mathrm{Up}4\left(X^{4}\right)+X^{3}$

Stage 3 to 2: $F_{up3}=\mathrm{Up}3\left(F_{up4}\right)+X^{2}$

Stage 2 to 1: $F_{up2}=\mathrm{Up}2\left(F_{up3}\right)+X^{1}$

Stage 1 to 0: $F_{up1}=\mathrm{Up}1\left(F_{up2}\right)+X^{0}$

Where $X^{i}$ represents the fused features at stage *i* obtained from the DAMFFM-HA fusion modules.

**Final Reconstruction.** The final reconstruction stage comprises a final upsampling block and an output convolutional layer. The upsampled features are enhanced through residual connections with the original input images:$$F_{final}=\mathrm{Up}0\left(F_{up1}\right)+I_{1}+I_{2}$$
where $I_{1}$ and $I_{2}$ are the original input images. The final output is obtained through a 3×3 convolution followed by a Sigmoid activation function, ensuring output values are constrained within the [0,1] range:$$I_{fused}=\sigma \left({\mathrm{Conv}}_{3\times 3}\left(F_{final}\right)\right)$$

This decoder design ensures the effective reconstruction of high-quality fused images while maintaining both global contextual information captured by the encoder and local spatial details preserved through skip connections with fused multi-scale features.

### 3.5. Loss Function

To effectively train our MFF network, we design a comprehensive loss function that combines multiple complementary loss terms to optimize the quality of fused images. The overall loss function is formulated as:(1)$$L_{total}=\lambda_{mse}L_{MSE}+\lambda_{ssim}L_{SSIM}+\lambda_{grad}L_{grad}$$
where $\lambda_{mse}$, $\lambda_{ssim}$, and $\lambda_{grad}$ represent the weighting coefficients for the mean squared error loss, structural similarity loss, and gradient loss, respectively.

The MSE loss measures pixel-wise differences between the fused image and the reference image, ensuring basic reconstruction fidelity:(2)$$L_{MSE}=\frac{1}{N}\sum_{i=1}^{N}{\left(I_{fused}\left(i\right)-I_{ref}\left(i\right)\right)}^{2}$$
where $I_{fused}$ denotes the fused image, $I_{ref}$ represents the reference image, and *N* is the total number of pixels.

The Structural Similarity Index Measure (SSIM) better captures perceptual image quality from the perspective of human visual perception. We employ 1−SSIM as our structural similarity loss:(3)$$L_{SSIM}=1-SSIM\left(I_{fused},I_{ref}\right)$$

The SSIM metric is computed as:(4)$$SSIM\left(x,y\right)=\frac{\left(2\mu_{x}\mu_{y}+C_{1}\right)\left(2\sigma_{xy}+C_{2}\right)}{\left(\mu_{x}^{2}+\mu_{y}^{2}+C_{1}\right)\left(\sigma_{x}^{2}+\sigma_{y}^{2}+C_{2}\right)}$$
where $\mu_{x}$ and $\mu_{y}$ are the local means of images *x* and *y*, $\sigma_{x}^{2}$ and $\sigma_{y}^{2}$ are the local variances, $\sigma_{xy}$ is the local covariance, and $C_{1}$ and $C_{2}$ are small constants for numerical stability.

To compute local statistics, we employ an 11×11 Gaussian weighting window with weights determined by:(5)$$w\left(x\right)=\exp\left(-\frac{{\left(x-\mu_{w}\right)}^{2}}{2\sigma_{w}^{2}}\right)$$
where $\sigma_{w}=1.5$ and $\mu_{w}$ corresponds to the window center. The weights are normalized to sum to unity.

The gradient loss preserves edge information and fine-grained textures in the fused image. We utilize Sobel operators to compute image gradients:(6)$$G_{x}=\left[\begin{array}{ccc} -1 & 0 & 1\\ -2 & 0 & 2\\ -1 & 0 & 1 \end{array}\right],\;G_{y}=\left[\begin{array}{ccc} 1 & 2 & 1\\ 0 & 0 & 0\\ -1 & -2 & -1 \end{array}\right]$$

The gradient magnitude is calculated as:(7)$$\nabla I=|G_{x}*I|\;+\;|G_{y}*I|$$
where ∗ denotes the convolution operation.

For MFF, the ideal fused image should preserve the strongest gradient information from the input images. Therefore, we define the joint gradient as:(8)$$\nabla I_{joint}=\max\left(\nabla I_{A},\nabla I_{B}\right)$$

The gradient loss is computed using the L1 norm:(9)$$L_{grad}=\frac{1}{N}\sum_{i=1}^{N}\left|\nabla I_{fused}\left(i\right)-\nabla I_{joint}\left(i\right)\right|$$

Through extensive empirical evaluation, we set the loss weights as: $\lambda_{mse}=0.2$, $\lambda_{ssim}=0.7$, and $\lambda_{grad}=0.1$. This configuration ensures that structural similarity plays a dominant role in the training process while maintaining appropriate pixel-level constraints and edge preservation capabilities. The emphasis on SSIM aligns with human visual perception, leading to perceptually superior fusion results.

## 4. Experiments

### 4.1. Experimental Setup

**Training Strategy.** We trained our model using the DUTS dataset (accessed on 3 July 2025) [61], which provides 15,572 images. We allocated 10,553 images for training and 5019 for validation, resizing all to 256×256 pixels. To create multi-focus image pairs, we transformed ground truth annotations into binary masks. These masks then guided the application of Gaussian blur, with kernel sizes from 3 to 21, to generate realistic training samples (detailed in SwinMFF [21]). For network optimization, we utilized the AdamW optimizer with an initial learning rate of $1\times 10^{-3}$, β parameters set to (0.9,0.999), an ϵ of $1\times 10^{-8}$, and a weight decay of 0.0001. A CosineAnnealingLR scheduler was implemented to progressively reduce the learning rate over the course of training. The model was trained for 20 epochs with a batch size of 16 on an Nvidia A6000 GPU system. operating at 2.90 GHz. The entire framework was developed in PyTorch 2.6.0.

**Datasets for Evaluation.** MLP-MFF’s performance was rigorously evaluated against three prominent MFF benchmarks: Lytro [28], MFFW [62], and MFI-WHU [41]. The Lytro dataset, consisting of 20 light-field camera image pairs, facilitated both qualitative and quantitative analysis. MFFW, with its 13 image pairs exhibiting pronounced defocus, was employed for qualitative evaluation. Similarly, the MFI-WHU dataset, offering 120 synthetically generated (via Gaussian blur), was also used for qualitative assessment.

**Methods for Comparison.** To comprehensively evaluate the performance of our proposed MLP-MFF, we compare it with various state-of-the-art methods from different categories. For traditional methods, we include both spatial domain and transform domain approaches. The spatial domain methods include SSSDI [63], QUADTREE [64], DSIFT [65], SRCF [28], GFDF [66], BRW [67], MISF [68], and MDLSR_RFM [69]. The transform domain methods include DWT [14], DTCWT [70], NSCT [71], CVT [72], GFF [73], SR [74], ASR [75], and ICA [76]. For decision map-based deep learning methods, we compare with CNN [19], ECNN [77], DRPL [78], SESF [31], MFIF-GAN [33], MSFIN [20], GACN [79], and ZMFF [35]. For end-to-end deep learning methods, we compare with IFCNN-MAX [39], U2Fusion [40], MFF-GAN [41], SwinFusion [22], FusionDiff [23], SwinMFF [21], and DDBFusion [42]. These methods represent the current state-of-the-art in MFF, covering different technical approaches and architectural designs. The comparison with these methods allows us to thoroughly evaluate the effectiveness and advantages of our proposed MLP-MFF approach.

**Evaluation metrics.** To comprehensively evaluate the performance of different fusion methods, we employ eight widely-used metrics that can be categorized based on their theoretical foundations. Information theory-based metrics include Entropy (EN) and Mutual Information (MI), which measure the information content and transfer in the fused image. Edge and gradient-based metrics consist of Edge Information (EI), Spatial Frequency (SF), Average Gradient (AVG), and $Q^{ab/f}$, which assess the preservation of edge details and image clarity. Structure and visual quality-based metrics include Structural Similarity Index Measure (SSIM) and Visual Information Fidelity (VIF), which evaluate the structural preservation and visual quality of the fused image. These metrics provide a comprehensive evaluation framework that considers different aspects of fusion quality, including information content, edge preservation, structural similarity, and visual quality.

### 4.2. Experimental Results

**Qualitative comparison.** In Figure 5, we present a comparative analysis of various MFF methods applied to the Lytro dataset [28]. The figure is structured into four rows, each representing a distinct category of fusion approaches: transform-domain methods, spatial-domain methods, decision map-based deep learning methods, and end-to-end deep learning methods. From the first example, it is evident that transform-domain methods and end-to-end deep learning methods consistently outperform others in preserving intricate details within complex scenes, exhibiting significantly fewer fusion artifacts. Furthermore, the second example clearly highlights the superior performance of our proposed method over existing end-to-end deep learning approaches. Our method effectively suppresses artifacts along object edges, a common challenge for other end-to-end techniques that show varying degrees of artifacts in the provided examples.

To provide a more intuitive comparison of the performance of different end-to-end methods, we further used the difference maps between the fusion results and the two source images as a basis for comparison. A greater discrepancy between the two difference maps indicates a superior fusion result [21]. First, we compare our method with decision map-based deep learning approaches in Figure 6. Even though our method is an end-to-end fusion approach where fused image pixel values are inferred by the network rather than sampled from source images, its difference maps are comparable to those produced by decision map-based deep learning methods. This demonstrates that our proposed method, similar to SwinMFF [21], achieves excellent pixel-level fidelity. Next, we compare our method with other end-to-end approaches in Figure 7. Our method, along with IFCNN-MAX and SwinMFF, significantly outperforms other methods in this comparison. While IFCNN-MAX and SwinMFF show slightly superior results to our proposed method, their computational costs are approximately 16 and 43 times higher, respectively. Therefore, our method strikes a favorable balance between computational efficiency and fusion quality.

To further evaluate the performance of different end-to-end methods in scenarios with strong defocus, we conducted an additional comparative analysis on the MFFW dataset [62], as shown in Figure 8. The results indicate that some methods, such as MFF-GAN [41] and DDBFusion [42], exhibit noticeable fusion errors in strongly defocused scenes. In contrast, our proposed method consistently maintains top-tier fusion quality across all examples, with virtually no significant artifacts appearing at the edges. Similarly, we used difference maps for further comparison in Figure 9. It is evident from the difference maps that several methods, including MFF-GAN [41], DDBFusion [42], and SwinFusion [22], produce two difference maps that are quite similar. This suggests that their fusion results fail to adequately distinguish and fuse the foreground and background in strongly defocused scenes. However, our proposed method, along with SwinMFF [21] and IFCNN-MAX [39], demonstrates significantly superior performance in these challenging conditions.

In Figure 10, we further visualize the fusion results of various end-to-end deep learning methods on the MFI-WHU dataset [41], along with their corresponding difference plots. The MFI-WHU dataset offers a more diverse range of examples, including small objects and high-contrast scenes, as depicted in Figure 10. The first example in Figure 10 demonstrates that our proposed method maintains robustness even with small objects, an area where many other methods, such as DDBFusion [42] and U2Fusion [40], are susceptible to noise and artifacts. Interestingly, while the diffusion model-based FusionDiff [23] generally exhibits poor performance and often produces noticeable color casts in previous examples, it performs exceptionally well in high-contrast scenes.

In terms of comprehensive performance across multiple datasets, our proposed method consistently delivers high-quality fusion results. It effectively handles complex scenarios, preserves fine details, and maintains excellent pixel-level fidelity. Furthermore, it demonstrates a strong ability to distinguish and fuse foreground and background elements, even in challenging conditions like strong defocus.

**Quantitative comparison.** Table 1 presents a comprehensive quantitative comparison of different MFF methods on the Lytro dataset [28]. The best-performing method for each metric is shown in bold, while the second-best is underlined. Additionally, a colored background is used to highlight the proposed method. An upward-pointing arrow (↑) next to a metric’s name indicates that a higher value is better. This formatting convention is applied consistently across all tables in this paper. The results demonstrate that our proposed MLP-MFF achieves superior performance across multiple evaluation metrics. Specifically, MLP-MFF achieves the highest scores in six out of eight metrics, significantly outperforming other methods in these aspects. End-to-end deep learning methods show varying performance levels. While DDBFusion achieves the highest SSIM score of 0.8661, it performs poorly in other metrics such as EI (48.1600) and SF (12.1484). SwinMFF shows balanced performance across multiple metrics but still falls short of MLP-MFF’s comprehensive superiority.

In Figure 11, we present the quantitative metrics for the fusion results of different methods on each image pair within the Lytro dataset [28]. The red line represents our proposed method. The results clearly show that the quantitative advantage of our proposed method extends across nearly the entire dataset, rather than being concentrated in just a few isolated examples that might skew overall metrics. This strong performance across diverse examples indicates that our proposed method possesses good generalization capabilities and superior performance.

In Table 2 and Table 3, we provide a further quantitative comparison of various deep learning-based methods on the MFFW dataset [62] and the MFI-WHU dataset [41], respectively. Across both datasets, our proposed method consistently ranks first or second in multiple metrics. This further demonstrates that our method maintains excellent fusion performance across a variety of scenarios.

The comprehensive quantitative experimental results collectively validate the superiority of the proposed MLP-MFF method across multiple evaluation metrics. MLP-MFF consistently demonstrates outstanding performance in terms of information entropy, edge information, structural similarity, and visual information fidelity, significantly outperforming both traditional methods and various existing deep learning approaches. Furthermore, MLP-MFF’s stable performance across different public datasets further underscores its strong generalization capability and robustness.

**Efficiency Analysis.** To comprehensively evaluate the computational efficiency of our proposed MLP-MFF, we compare it with various state-of-the-art learning-based MFF methods in terms of model size, computational complexity (FLOPs), and inference time. Note that all FLOPs are calculated on 256×256 input images to ensure fair comparison, and the inference time is measured as the average processing time per image on the MFI-WHU dataset [41]. As shown in Table 4, MLP-MFF demonstrates remarkable efficiency advantages across multiple dimensions. In terms of computational complexity, MLP-MFF achieves the lowest FLOPs (0.52G) among all compared methods, representing an 83.12% reduction compared to the previous most efficient method (MFF-GAN with 3.08G FLOPs). Regarding inference speed, MLP-MFF achieves the fastest inference time (0.01s), which is 83.33% faster than the previous fastest method (MFF-GAN with 0.06s). While MLP-MFF’s model size (1.23M) is not the smallest among all methods, it remains highly competitive, especially when considering its superior fusion performance demonstrated in Table 1. The model size is significantly smaller than recent Transformer-based methods such as SwinMFF (41.25M), making it more practical for deployment in resource-constrained environments. The results demonstrate that MLP-MFF successfully achieves an optimal balance between computational efficiency and fusion performance, making it a practical solution for real-world applications.

### 4.3. Performance Under Extreme Situations

To evaluate the robustness of the proposed MLP-MFF model, we performed a challenging experiment using images from the MFI-WHU dataset [41]. We randomly selected three images (as depicted in Figure 12)—and synthesized a corresponding fully blurred version of each using a Gaussian blur filter. This process effectively simulates severe defocus, such as that caused by camera shake or drastically incorrect focus settings. Even when one source image was completely blurred, our model demonstrated a remarkable ability to produce a fully clear output image.

### 4.4. Performance on Challenging Scenarios

To further assess our model’s robustness, we evaluated its performance on two challenging scenarios: high-contrast scenes and scenes containing small objects. As shown in Figure 13, the first row demonstrates our model’s effectiveness on a high-contrast scene, where the dark background and bright sign create significant variations. Our method successfully merges the focused regions without introducing artifacts or halo effects. The second row illustrates the model’s ability to handle small, intricate details, such as the waving flag. The fused image accurately preserves the fine textures of the flag, demonstrating the model’s capacity to maintain detail even in complex scenarios. These results highlight the model’s general applicability and robustness to diverse image characteristics.

### 4.5. Performance in Real-World Scenarios

As demonstrated by the fusion results on the Road-MF dataset [3] in Figure 14, MFF can serve as a crucial preprocessing step to significantly enhance the safety of autonomous driving systems. By fusing multiple images captured with different focal settings, MFF effectively extends the depth of field, ensuring that both near and far objects on the road, such as pedestrians, vehicles, and traffic signs, are simultaneously in sharp focus. This improved clarity provides downstream perception modules with more reliable and detailed visual information. Consequently, this leads to more robust and accurate environmental perception, which is essential for making timely and safe navigation decisions in complex real-world driving conditions. Furthermore, the proposed method can achieve this fusion with a nearly negligible computational overhead and time cost on current automotive-grade processors.

### 4.6. Performance of Processing Image Sequences

To demonstrate the scalability of our method for fusing more than two images, we applied it to four image sequences from the Lytro dataset [28]. We adopted a sequential fusion strategy: first, we fused the initial two source images, and then we fused this intermediate result with the third image to generate the final output.

As shown in Figure 15, the fused images successfully preserve all the in-focus regions from the multiple inputs, resulting in a comprehensive, all-in-focus image with excellent visual quality. This experiment confirms that our method can be effectively extended to handle multiple source images by applying it in a sequential manner.

### 4.7. Ablation Study

To validate the effectiveness of our proposed DAMFFM-HA module, we conducted an ablation study on the Lytro dataset, with results presented in Table 5. As observed, when the DAMFFM-HA module is removed and the pixel-wise dot product is directly used as the fusion scheme, the model experiences a performance decline across various metrics. Conversely, incorporating DAMFFM-HA leads to a significant improvement in all metrics, demonstrating the module’s substantial role in enhancing the fused image’s edge information, structural similarity, and visual information fidelity.

Additionally, we compared the performance of different backbone network architectures for end-to-end MFF. This comparison included UniRepLKNet [80], representing one of the most advanced CNN networks; Swin Transformer [22], representing Transformer-based networks; EVMamba [81], representing vision state space-based networks; and MAXIM [59], another widely used MLP architecture. The results, shown in Table 6, reveal that MLP-MFF, which employs PFMLP as its backbone, achieves optimal results on most metrics compared to other mainstream architectures. This further validates the effectiveness and superiority of PFMLP in MFF tasks.

To further address the architectural design choices, particularly the number of blocks within each stage, we conducted a comparison with various configurations from the original PFMLP paper. This approach, as adopted by the original PFMLP work, is a more common and effective method to analyze the influence of network depth than varying the number of stages. As shown in Table 7, we compare our model (“PFMLP-Ours”) with four versions of the PFMLP backbone: PFMLP-N, PFMLP-T, PFMLP-S, and PFMLP-B. The results indicate that while increasing the number of blocks (from N to T, S, and B) leads to a marginal improvement in fusion results, this gain comes with a significant increase in model size (Params) and computational cost (FLOPs). For instance, the PFMLP-B model achieves the highest scores but with a substantial increase in complexity. Our model, which utilizes a single block per stage, achieves a satisfactory balance between performance and efficiency, demonstrating the rationality of our architectural choice.

In summary, the ablation experiments conclusively demonstrate the significant contribution of both the proposed DAMFFM-HA module and the PFMLP backbone network in enhancing MFF performance.

## 5. Discussion

In this study, we explored a MLP-based architecture for MFF. Despite its strengths, our model has two key limitations. First, it relies on synthetically generated defocused images for training, which may not fully represent the complexities of real-world defocus patterns. Second, the architecture is currently optimized for dual-source fusion and would require significant modification to be extended to multi-source or multi-modal scenarios. In the future, we plan to address these limitations by exploring more advanced training techniques to improve performance on real-world data and investigating how to extend this architecture to handle multi-focus image stack fusion.

## 6. Conclusions

This paper introduces MLP-MFF, a novel lightweight end-to-end MFF network built upon the Pyramid Fusion Multi-Layer Perceptron (PFMLP) architecture. Our method directly addresses common limitations of existing deep learning multi-focus fusion approaches, such as high computational complexity, inflexible input sizes, and restricted global receptive fields. We leverage the inherent strengths of the PFMLP backbone and introduce a Dual-path Adaptive Multi-scale Feature-Fusion Module with Hybrid Attention (DAMFFM-HA), which effectively models both local and global dependencies and adaptively integrates multi-scale features.

Extensive experiments on multiple public MFF datasets demonstrate that MLP-MFF meets or exceeds the fusion quality of current mainstream methods, all while maintaining exceptional efficiency and a lightweight design. Furthermore, MLP-MFF significantly reduces computational complexity and inference time, making it highly suitable for practical applications, especially in resource-constrained environments.

## Figures and Tables

**Figure 1 sensors-25-05146-f001:**
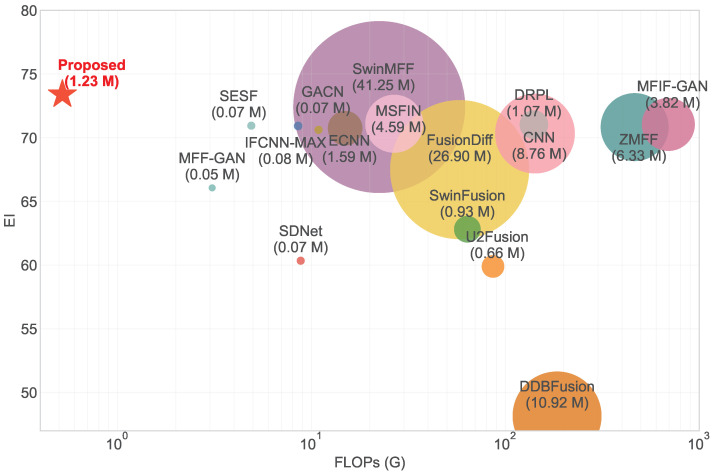
Quantitative comparison of model size (indicated by bubble size), computational complexity (FLOPs), and fusion quality (EI) among different deep learning-based MFF methods on the Lytro dataset [28].

**Figure 2 sensors-25-05146-f002:**
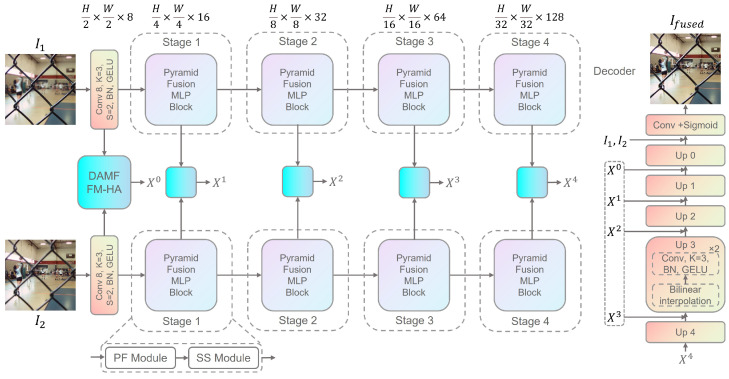
The framework of the proposed MLP-MFF.

**Figure 3 sensors-25-05146-f003:**
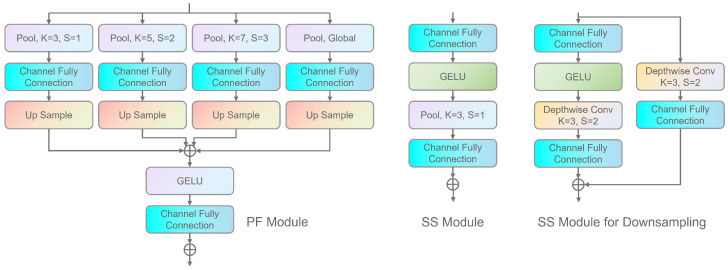
The details of PF Module and SS Module.

**Figure 4 sensors-25-05146-f004:**
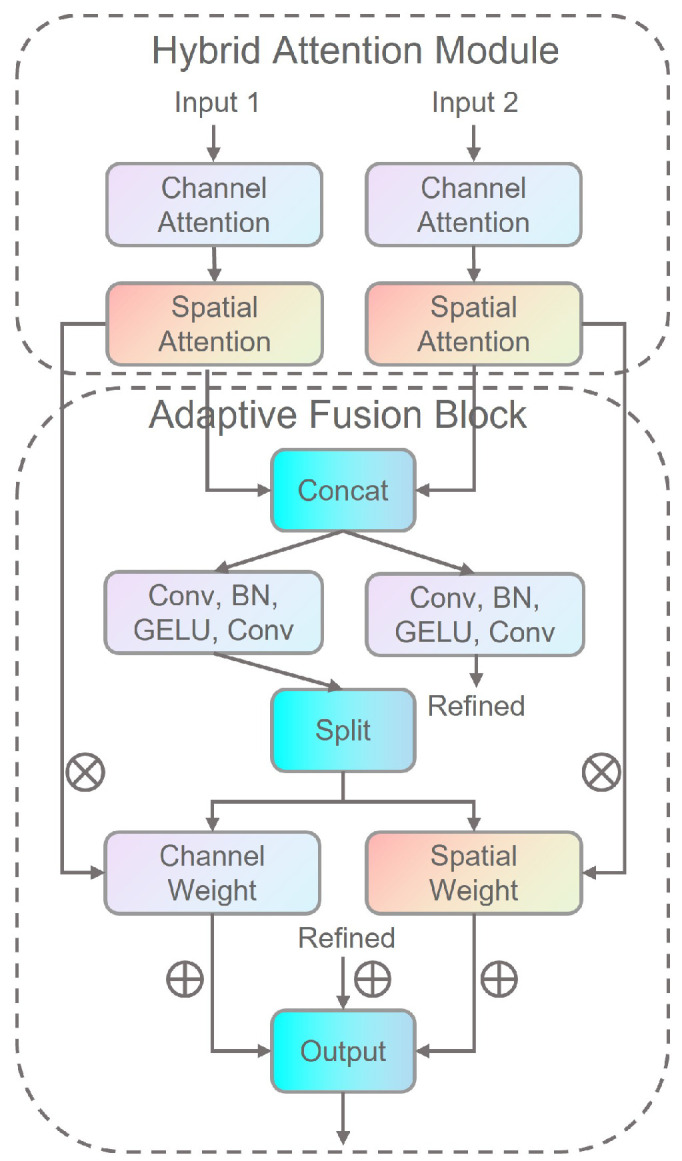
The details of the dual-path adaptive multi-scale feature-fusion module based on hybrid attention (DAMFFM-HA).

**Figure 5 sensors-25-05146-f005:**
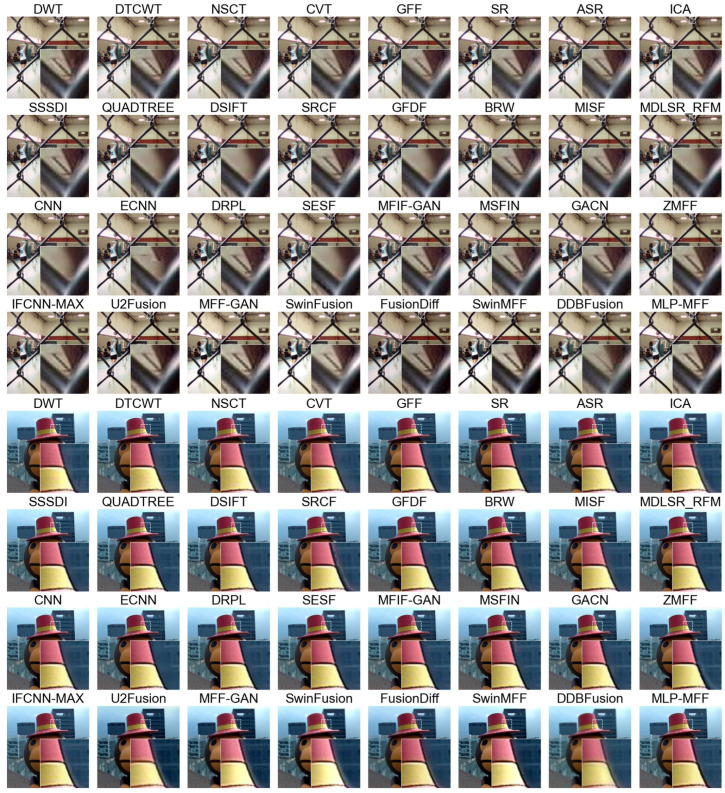
The fusion results of various SOTA methods on the Lytro dataset [28].

**Figure 6 sensors-25-05146-f006:**
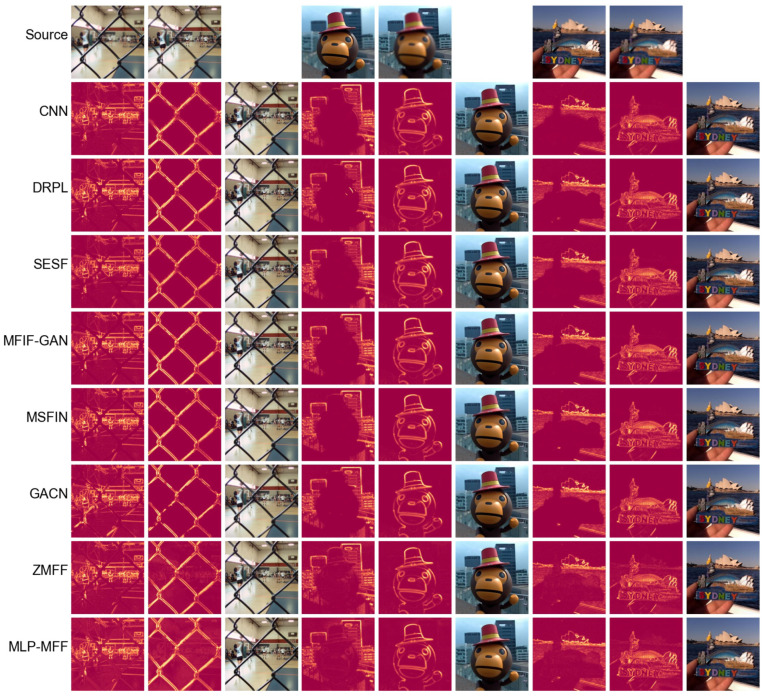
The difference maps of various SOTA decision map-based MFF methods implemented using deep learning on the Lytro dataset [28].

**Figure 7 sensors-25-05146-f007:**
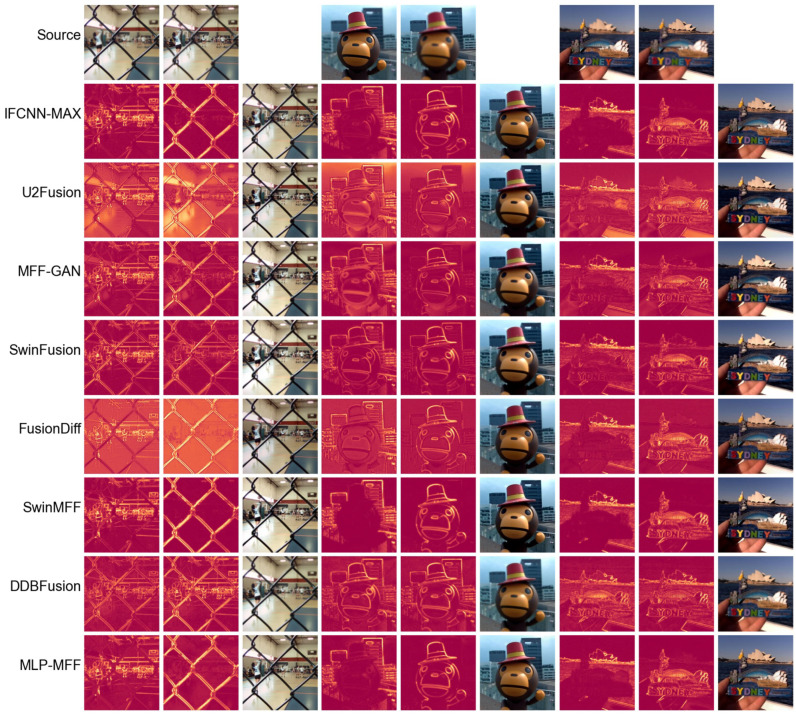
The difference maps of different SOTA end-to-end MFF methods implemented using deep learning on the Lytro dataset [28].

**Figure 8 sensors-25-05146-f008:**
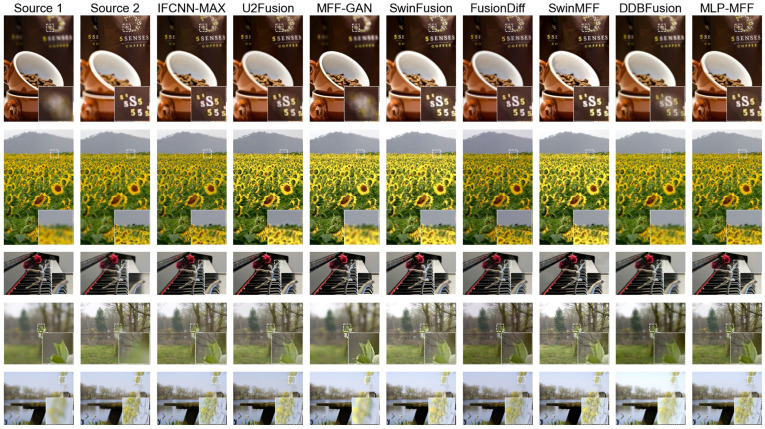
The fusion results of various SOTA methods on the MFFW dataset [62].

**Figure 9 sensors-25-05146-f009:**
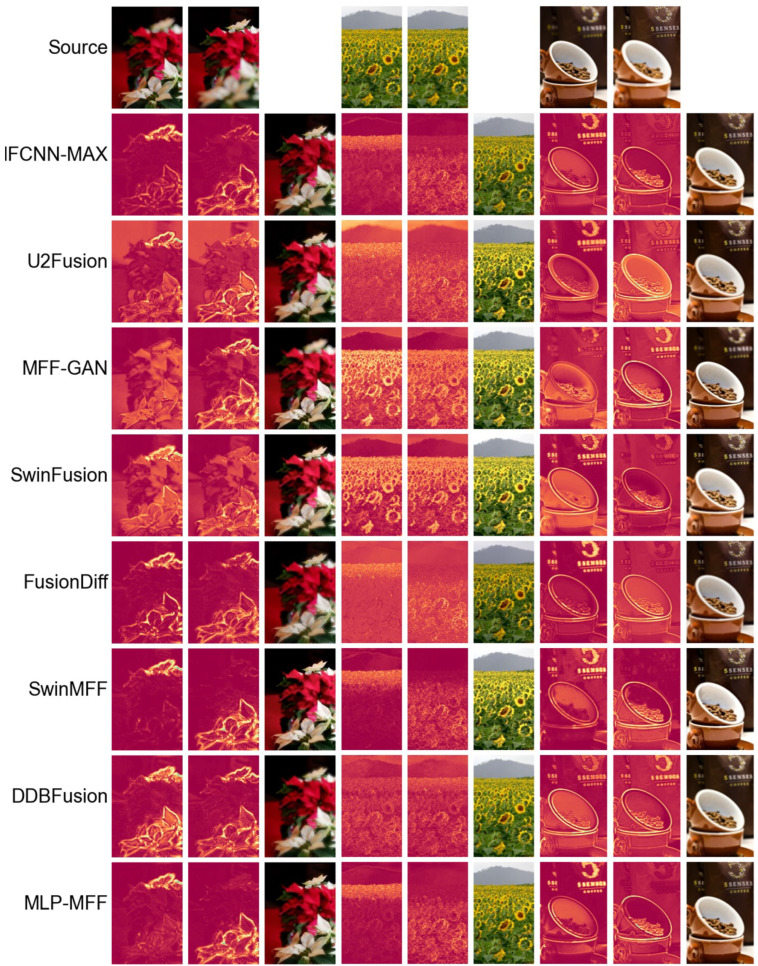
The difference maps of different SOTA end-to-end MFF methods implemented using deep learning on the MFFW dataset [62].

**Figure 10 sensors-25-05146-f010:**
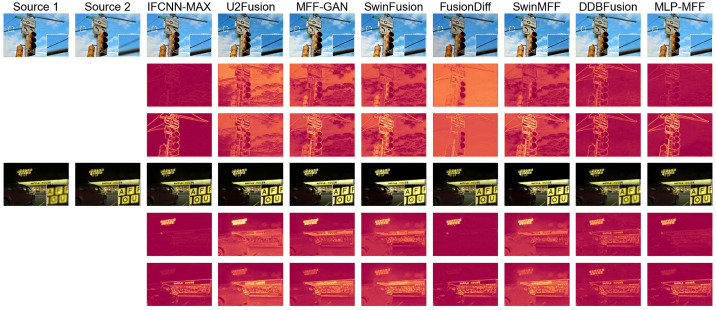
The fusion results of various SOTA methods on the MFI-WHU dataset [41].

**Figure 11 sensors-25-05146-f011:**
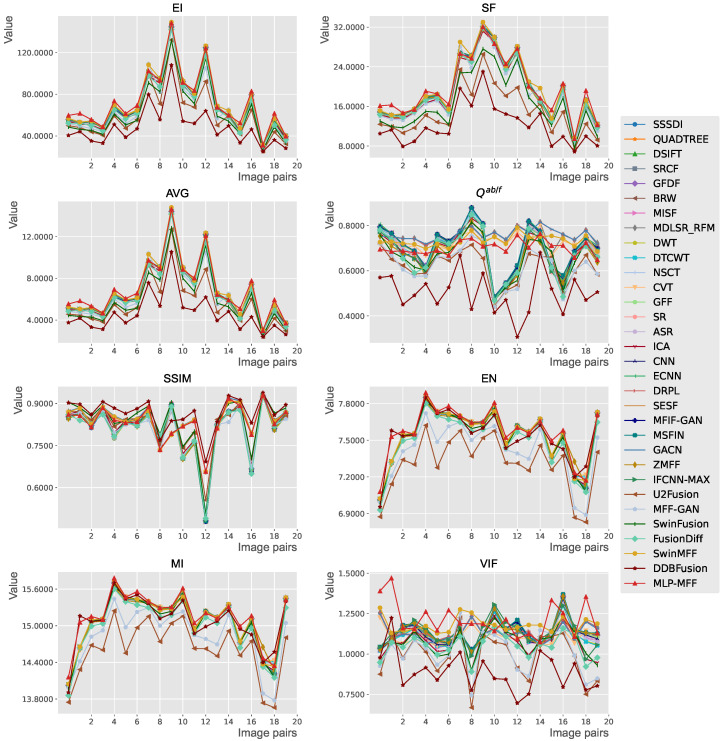
Objective performance of different fusion methods on the Lytro [28] dataset.

**Figure 12 sensors-25-05146-f012:**
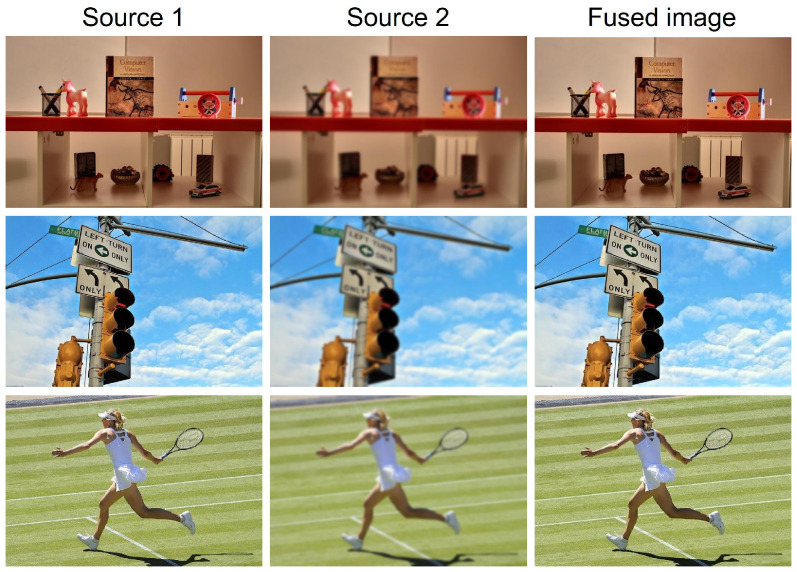
Performance evaluation under extreme conditions. From left to right: original sharp image, synthetically blurred counterpart, and final fusion result.

**Figure 13 sensors-25-05146-f013:**
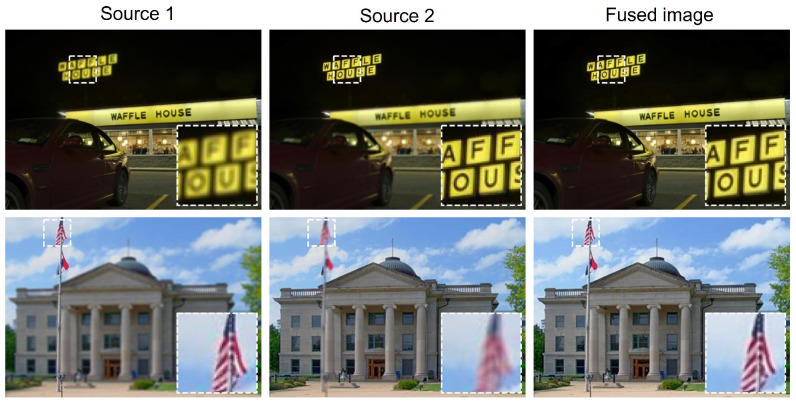
Fusion results on challenging scenarios: the first row shows a high-contrast scene, and the second row shows a scene with small objects.

**Figure 14 sensors-25-05146-f014:**
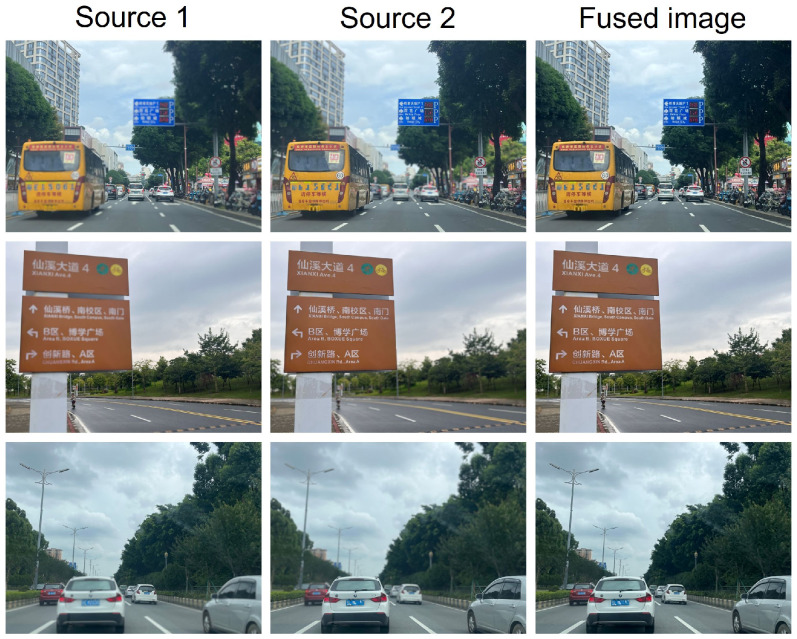
Fusion results on Road-MF [3] dataset.

**Figure 15 sensors-25-05146-f015:**
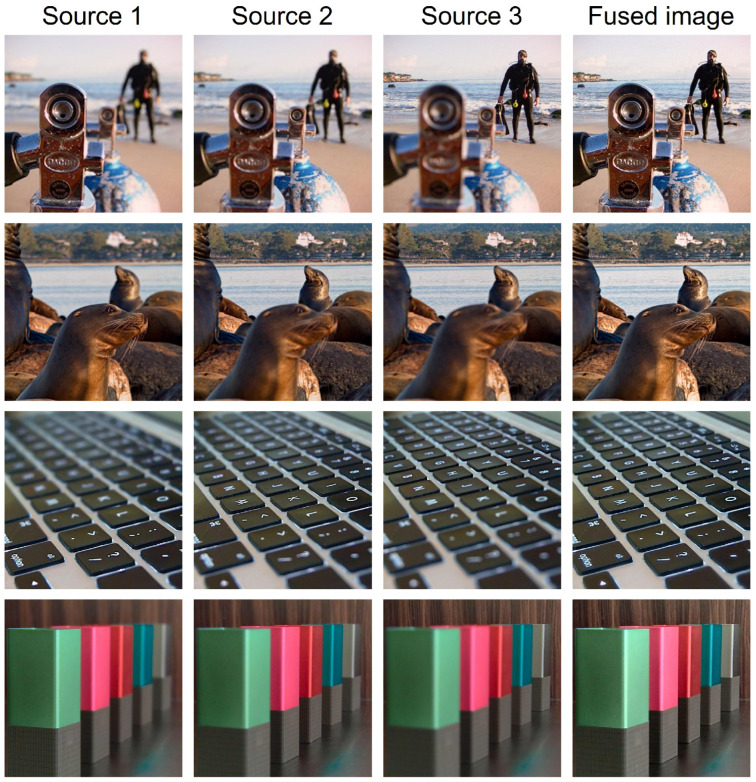
Consecutive multi-focus image-fusion results on Lytro [28] triplet dataset.

**Table 1 sensors-25-05146-t001:** Quantitative comparison of different MFF methods on the Lytro dataset [28].

Methods	EI ↑	SF↑	AVG↑	$Q^{\mathrm{ab}/f}$↑	SSIM↑	EN↑	MI↑	VIF↑
*Methods based on image spatial domain*								
SSSDI [63]	70.7102	19.3567	6.8234	0.6966	0.8069	7.5334	15.0668	1.1309
QUADTREE [64]	70.8957	19.4163	6.8412	0.7027	0.8085	7.5342	15.0684	1.1368
DSIFT [65]	70.9808	19.4194	6.8493	0.7046	0.8083	7.5344	15.0688	1.1381
SRCF [28]	71.0810	19.4460	6.8607	0.7036	0.8075	7.5345	15.0690	1.1374
GFDF [66]	70.6258	19.3312	6.8145	0.7049	0.8098	7.5337	15.0674	1.1336
BRW [67]	70.6777	19.3433	6.8200	0.7040	0.8093	7.5337	15.0675	1.1336
MISF [68]	70.4148	19.2203	6.7945	0.6984	0.8084	7.5335	15.0671	1.1222
MDLSR_RFM [69]	70.9078	19.4100	6.8422	0.7518	0.8393	7.5343	15.0686	1.1353
*Methods based on image transform domain*								
DWT [14]	70.7942	19.3342	6.8336	0.6850	0.8059	7.5436	15.0872	1.1114
DTCWT [70]	70.5666	19.3204	6.8134	0.6929	0.8076	7.5396	15.0791	1.1079
NSCT [71]	70.4289	19.2662	6.8027	0.6901	0.8102	7.5408	15.0816	1.1249
CVT [72]	70.3233	19.2713	6.7897	0.7243	0.8376	7.5414	15.0828	1.1044
GFF [73]	70.5179	19.2947	6.8058	0.6998	0.8088	7.5358	15.0716	1.1277
SR [74]	70.2498	19.2819	6.7818	0.6944	0.8097	7.5325	15.0650	1.1208
ASR [75]	70.3342	19.2818	6.7897	0.6951	0.8093	7.5327	15.0654	1.1201
ICA [76]	68.3180	18.5968	6.6125	0.6766	0.8176	7.5327	15.0655	1.0708
*Decision map-based methods using deep learning*								
CNN [19]	70.3238	19.2295	6.7860	0.7019	0.8096	7.5331	15.0663	1.1255
ECNN [77]	70.7432	19.3837	6.8261	0.7030	0.8089	7.5338	15.0675	1.1337
DRPL [78]	71.0214	19.4546	6.8531	0.7574	0.8401	7.5342	15.0683	1.1393
SESF [31]	70.9403	19.4158	6.8448	0.7031	0.8086	7.5348	15.0696	1.1395
MFIF-GAN [33]	71.0395	19.4370	6.8560	0.7029	0.8078	7.5345	15.0690	1.1393
MSFIN [20]	71.0914	19.4438	6.8602	0.7045	0.8082	7.5348	15.0695	1.1420
GACN [79]	70.6148	19.3087	6.8101	**0.7581**	0.8413	7.5330	15.0661	1.1304
ZMFF [35]	70.8298	18.9707	6.8045	0.6635	0.8073	7.5368	15.0735	1.1331
*End-to-end methods based on deep learning*								
IFCNN-MAX [39]	70.9193	19.3793	6.8463	0.6784	0.8111	7.5361	15.0722	1.1322
U2Fusion [40]	59.8957	14.9334	5.6515	0.6190	0.8239	7.3077	14.6153	0.9882
MFF-GAN [41]	66.0601	18.4022	6.4089	0.6222	0.8067	7.4076	14.8153	1.0084
SwinFusion [22]	62.8130	16.6430	5.9862	0.6597	0.8367	7.5238	15.0476	1.0685
FusionDiff [23]	67.4911	18.8483	6.5325	0.6744	0.8071	7.4909	14.9817	1.0448
SwinMFF [21]	72.4041	19.7954	6.9734	0.7321	0.8382	7.5413	15.0826	1.1810
DDBFusion [42]	48.1600	12.1484	4.5883	0.5026	**0.8661**	7.5332	15.0663	0.8874
MLP-MFF	**73.3902**	**19.8181**	**7.0501**	0.7025	0.8344	**7.5741**	**15.1482**	**1.2126**

**Table 2 sensors-25-05146-t002:** Quantitative comparison of different MFF methods on the MFFW dataset [62].

Methods	EI ↑	SF↑	AVG↑	$Q^{\mathrm{ab}/f}$↑	SSIM↑	EN↑	MI↑	VIF↑
*Decision map-based deep learning methods*								
CNN [19]	73.9624	22.1315	7.4426	0.6216	0.8071	7.1749	14.3497	1.0305
ECNN [77]	75.7644	22.7149	7.6464	**0.7312**	0.8136	7.1904	14.3807	1.0541
DRPL [78]	77.1437	23.2078	7.8088	0.7228	0.8101	7.1944	14.3889	1.0552
SESF [31]	76.9227	23.1558	7.7519	0.6247	0.7947	7.1920	14.3841	1.0542
MFIF-GAN [33]	76.5417	22.9481	7.7225	0.7283	0.8138	7.1570	14.3140	1.0639
MSFIN [20]	75.7969	22.8168	7.6375	0.6183	0.8007	7.1764	14.3528	1.0616
GACN [79]	75.7403	22.5541	7.6327	0.7293	0.8162	7.1951	14.3902	1.0449
ZMFF [35]	77.7055	21.4789	7.6592	0.6541	0.8110	7.1665	14.3329	1.0636
*End-to-end methods based on deep learning*								
IFCNN-MAX [39]	76.3056	22.1333	7.6334	0.6022	0.8152	7.1710	14.3420	1.0344
U2Fusion [40]	65.7906	16.6017	6.3099	0.5992	0.8178	6.9057	13.8115	0.9189
MFF-GAN [41]	**83.0560**	**28.2025**	**8.4157**	0.4372	0.7482	7.1731	14.3462	**1.2342**
SwinFusion [22]	75.3649	20.5358	7.3528	0.6423	0.8102	7.1419	14.2838	1.1912
FusionDiff [23]	69.6123	21.2969	7.0366	0.6673	0.8198	7.1138	14.2275	0.9052
SwinMFF [21]	80.4903	22.7120	7.9646	0.6636	0.8198	7.1921	14.3843	1.1577
DDBFusion [42]	55.1218	14.7261	5.3471	0.4803	**0.8391**	7.1966	14.3932	0.8952
MLP-MFF	79.5011	22.0943	7.8559	0.6392	0.8222	**7.2152**	**14.4303**	1.1756

**Table 3 sensors-25-05146-t003:** Quantitative comparison of different MFF methods on the MFI-WHU dataset [41].

Methods	EI ↑	SF↑	AVG↑	$Q^{\mathrm{ab}/f}$↑	SSIM↑	EN↑	MI↑	VIF↑
*Decision map-based deep learning methods*								
CNN [19]	77.0123	26.4975	8.1720	0.7276	0.8310	7.3173	14.6345	1.0959
ECNN [77]	77.9532	26.7520	8.2718	**0.7314**	0.8296	7.3205	14.6411	1.1038
DRPL [78]	78.5301	**26.9109**	8.3340	0.7305	0.8298	7.3222	14.6443	1.1126
SESF [31]	77.6439	26.7527	8.2356	0.7267	0.8293	7.3202	14.6404	1.1078
MFIF-GAN [33]	78.5272	26.9048	8.3274	0.7302	0.8288	7.3247	14.6494	1.1169
MSFIN [20]	77.6764	26.8228	8.2380	0.7273	0.8296	7.3173	14.6345	1.1118
GACN [79]	76.8219	26.5318	8.1374	0.7259	0.8309	7.3141	14.6283	1.1027
ZMFF [35]	**82.0595**	24.9329	**8.3925**	0.6193	0.7947	7.2790	14.5580	1.1742
*End-to-end methods based on deep learning*								
IFCNN-MAX [39]	79.3862	26.6642	8.3756	0.6936	0.8325	7.3331	14.6662	1.1713
U2Fusion [40]	68.8453	18.1867	6.6806	0.5917	0.8413	7.1315	14.2630	1.1349
SDNet [62]	70.8002	24.2105	7.5039	0.6889	0.8478	7.2619	14.5238	1.0236
MU [62]	76.7037	25.2436	8.0277	0.6496	0.8320	7.2336	14.4673	1.1368
MFF-GAN [41]	68.6117	20.6637	6.9656	0.6777	**0.8620**	7.2894	14.5788	1.1060
SwinFusion [22]	79.1056	21.8127	7.8106	0.5080	0.7185	7.3771	14.7543	1.1136
FusionDiff [23]	72.3067	23.6592	7.5304	0.6762	0.8213	7.2758	14.5516	1.0399
SwinMFF [21]	78.9436	26.3398	8.3138	0.7008	0.8301	7.3274	14.6548	1.1379
DDBFusion [42]	56.5089	17.0766	5.7231	0.5102	0.8510	7.3151	14.6302	1.0139
MLP-MFF	80.9240	25.3620	8.3868	0.6673	0.8291	**7.3804**	**14.7607**	**1.2532**

**Table 4 sensors-25-05146-t004:** Comparison of computational efficiency across different learning-based MFF methods.

Method	Model Size (M)	FLOPs (G)	Time (s)
*Decision map-based methods using deep learning*			
CNN [19]	8.76	142.23	0.06
ECNN [77]	1.59	14.93	125.53
DRPL [78]	1.07	140.49	0.22
SESF [31]	0.07	4.90	0.26
MFIF-GAN [33]	3.82	693.03	0.32
MSFIN [20]	4.59	26.76	1.10
GACN [79]	0.07	10.89	0.16
ZMFF [35]	6.33	464.53	165.38
*End-to-end methods based on deep learning*			
IFCNN-MAX [39]	0.08	8.54	0.09
U2Fusion [40]	0.66	86.40	0.16
MFF-GAN [41]	**0.05**	3.08	0.06
SwinFusion [22]	0.93	63.73	1.79
FusionDiff [23]	26.90	58.13	81.47
SwinMFF [21]	41.25	22.38	0.46
DDBFusion [42]	10.92	184.93	1.69
MLP-MFF	1.23	**0.52**	**0.01**
Reduction (%)	/	83.12%	83.33%

**Table 5 sensors-25-05146-t005:** Ablation study on the effectiveness of the proposed DAMFFM-HA module.

Settings	EI ↑	SF↑	AVG↑	$Q^{\mathrm{ab}/f}$↑	SSIM↑	EN↑	MI↑	VIF↑
w/o DAMFFM-HA	72.5682	19.0216	6.9557	0.6965	0.8253	7.5353	15.0660	1.1772
w DAMFFM-HA	**73.3902**	**19.8181**	**7.0501**	**0.7025**	**0.8344**	**7.5741**	**15.1482**	**1.2126**

**Table 6 sensors-25-05146-t006:** Comparison of different backbone architectures for end-to-end MFF.

Backbone	EI ↑	SF↑	AVG↑	$Q^{\mathrm{ab}/f}$↑	SSIM↑	EN↑	MI↑	VIF↑
UniRepLKNet [80]	73.0187	19.5731	6.9837	0.6987	0.8261	7.5381	15.0629	1.1716
Swin Trans. [82]	72.3936	19.5826	6.9265	**0.7215**	0.8288	7.5378	15.0593	1.1783
EVMamba [81]	72.3195	19.4956	6.9474	0.7021	0.8292	7.5351	15.0557	1.1673
MAXIM [59]	72.7428	19.6851	7.0185	0.6985	0.8302	7.5462	15.0926	1.1827
PFMLP-Ours	**73.3902**	**19.8181**	**7.0501**	0.7025	**0.8344**	**7.5741**	**15.1482**	**1.2126**

**Table 7 sensors-25-05146-t007:** Comparison with several model versions provided in the original PFMLP paper.

Backbone	EI↑	SF↑	AVG↑	$Q^{\mathrm{ab}/f}$↑	SSIM↑	EN↑	MI↑	VIF↑
PFMLP-Ours	73.3902	19.8181	7.0501	0.7025	0.8344	7.5741	15.1482	1.2126
PFMLP-N	73.3951	19.8223	7.0542	0.7029	0.8348	7.5780	15.1520	1.2130
PFMLP-T	73.3995	19.8265	7.0583	0.7033	0.8352	7.5819	15.1558	1.2134
PFMLP-S	73.4040	19.8307	7.0624	0.7037	0.8356	7.5858	15.1596	1.2138
PFMLP-B	**73.4085**	**19.8349**	**7.0665**	**0.7041**	**0.8360**	**7.5897**	**15.1634**	**1.2142**

## Data Availability

The code of this study will be publicly accessible in a GitHub repository at https://github.com/Xinzhe99/MLP-MFF (accessed on 3 July 2025).

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
