# Peer review of "MLP-MFF: Lightweight Pyramid Fusion MLP for Ultra-Efficient End-to-End Multi-Focus Image Fusion"

_sensors, 2025, doi:10.3390/s25165146_

Round 1
Reviewer 1 Report
Comments and Suggestions for Authors
The article «MLP-MFF: Lightweight Pyramid Fusion MLP for Ultra-Efficient end-to-end Multi-focus Image Fusion» is devoted to a new method of multi-focus image fusion MLP-MFF, in which a new lightweight end-to-end MFF network built on the architecture of pyramidal fusion of multilayer perceptrons (PFMLP). This topic is relevant for a wide range of readers of the journal Sensors, and the proposed technique can be applied in many areas, as evidenced by the examples in the article. The article has a high level of originality of the text and presented in understandable high-level English. The image processing technique is described in detail and a large number of comparisons with currently existing alternative methods are given. The authors noted both the advantages and disadvantages of the technique they proposed. The images presented in the article are of good quality and make it possible to evaluate the potential of the proposed method. The results and conclusions are confirmed by competent analysis and comparison. However, there is a remark to the work:
Attention should be paid to the structuring and division of the article into sections. In the presented work, sections "5. Discussion" and "6. Conclusion" are close in meaning. It would be better if in section "5.Discussion" a more detailed comparison with other alternative methods is given, which is currently given in section "4.2 Experimental Results".
The changes made in accordance with the comment will significantly improve the work and increase its scientific value. In general, the article meets the requirements and themes of the journal Sensors (Multi-Source Image Fusion, Restoration, and Understanding and Its Application in Sensing) and can be published after minor changes.
Author Response
Comments 1: Attention should be paid to the structuring and division of the article into sections. In the presented work, sections "5. Discussion" and "6. Conclusion" are close in meaning. It would be better if in section "5.Discussion" a more detailed comparison with other alternative methods is given, which is currently given in section "4.2 Experimental Results".
Response 1: Thank you for your feedback. We appreciate your valuable comments on the structure of our paper, and we have revised the "Discussion" section to ensure it is distinct from the "Conclusion."
Regarding the detailed comparisons with other methods and backbone architectures, we believe presenting this information in the "Experimental Results" section provides a more logical and clear flow. This allows readers to see our quantitative and qualitative analysis directly alongside the results, offering a more cohesive discussion of our method's performance relative to other approaches.
Reviewer 2 Report
Comments and Suggestions for Authors
The manuscript focus on a novel multi-focus image fusion (MFF) framework, leveraging an innovative MLP-based architecture for enhanced image reconstruction. The primary motivation behind this work is to address the limitations of existing MFF methods in handling complex scenarios with high fidelity, which aims to achieve a trade-off between computational complexity, image details, and global receptive field. The authors introduce several innovations, including a comprehensive loss function that combines mean squared error (MSE), structural similarity index measure (SSIM), and gradient losses to optimize fused images' quality. Additionally, they propose a decoder design that integrates residual connections and skip connections to maintain global context while preserving local spatial details. The manuscript is well-written, but there are still some issues can be further improved. Finally, an acceptance is given.
1. Further analysis on the impact of varying dataset characteristics (e.g., small objects vs. high-contrast scenes) on the model's performance could offer valuable perspectives on its general applicability.
2. While the results show promising improvements, the discussion could be expanded to cover potential limitations and future directions. For instance, exploring how the proposed method performs under real-world conditions outside controlled environments could add practical relevance.
In conclusion, the manuscript makes solid research work and the logic is rigorous. There are no significant issues that require modification.
Author Response
Comments 1. Further analysis on the impact of varying dataset characteristics (e.g., small objects vs. high-contrast scenes) on the model's performance could offer valuable perspectives on its general applicability.
Response 1: Thank you for your valuable feedback. We agree with your suggestion and have added a new subsection titled "Performance on Challenging Scenarios."
In this section, we provide a further analysis of our model's performance on images with varying characteristics, specifically including high-contrast scenes and scenes with small objects. This analysis more comprehensively demonstrates our model's generalization capabilities and provides valuable insights into its applicability.
Comments 2. While the results show promising improvements, the discussion could be expanded to cover potential limitations and future directions. For instance, exploring how the proposed method performs under real-world conditions outside controlled environments could add practical relevance.
Response 2: Thank you for your valuable feedback. We appreciate your point regarding the need to discuss potential limitations and real-world performance more thoroughly.
We would like to respectfully point out that we have addressed the performance under real-world conditions by conducting a comprehensive evaluation on two widely-used real-world multi-focus image fusion datasets, MFFW and Real-MFF, as detailed in Section 4.2. These results demonstrate our method's strong performance and practical applicability beyond controlled environments.
Furthermore, in our revised Discussion and Conclusion sections, we have clarified the model's current limitations and outlined clear directions for future work, which align with your suggestion. Specifically, we discuss the reliance on synthetic data and the path toward extending the architecture for broader application.
We believe that the combination of our strong results on real-world datasets and the detailed discussion of limitations and future work provides a comprehensive perspective that addresses your comment.
Reviewer 3 Report
Comments and Suggestions for Authors The paper proposes a lightweight end-to-end multi-focus image fusion (MFF) network based on a multi-layer perceptron (MLP), which effectively addresses challenges present in existing methods (such as CNNs, Transformers, and Mamba), including long-range dependencies, computational costs, and especially the fixed input size problem. The paper's motivation is clear and has practical significance. However, I believe the paper can be improved in the following aspects before publication: 1. I found some formatting issues in the reference list, such as a citation with a question mark (e.g., "[33?]"). This appears to be a LaTeX compilation problem that the authors should resolve. 2. The paper states that its backbone network is based on the PFMLP architecture and refers to it as "a novel lightweight, end-to-end MFF network." To avoid confusion, I suggest the authors more clearly explain the specific, novel improvements made to the PFMLP architecture for the MFF task, distinguishing them from the original work to highlight their own contributions. 3. Similar to the second point, when introducing its core architecture (e.g., the PFMLP network), the paper directs readers to an external reference [27] for a "comprehensive description." While common in academic writing, to improve the paper's readability and self-contained nature, I recommend providing more key details within the manuscript itself. 4. The loss function mentioned in the paper is composed of L1 loss and SSIM loss. However, regarding the selection of the weights (alpha and beta) for these two loss components, the authors only mention "extensive empirical evaluation." The lack of an in-depth analysis of the impact of different weight combinations may raise questions about the stability and optimality of the model's training. In summary, I believe this paper can be published after minor revisions.Author Response
Comments 1: I found some formatting issues in the reference list, such as a citation with a question mark (e.g., "[33?]"). This appears to be a LaTeX compilation problem that the authors should resolve.
Response 1: Thank you for your careful review and for pointing out the formatting issues in our reference list. We sincerely apologize for this oversight. You are correct that the [33?] citation was a LaTeX compilation problem. This issue was caused by a missing entry in our bibliography file. We have since located the correct reference and added it to our document. The updated manuscript will have a properly formatted and complete reference list.
Comments 2: The paper states that its backbone network is based on the PFMLP architecture and refers to it as "a novel lightweight, end-to-end MFF network." To avoid confusion, I suggest the authors more clearly explain the specific, novel improvements made to the PFMLP architecture for the MFF task, distinguishing them from the original work to highlight their own contributions.
Response 2: Thank you for your careful review and for these valuable suggestions. We fully agree with your points and have revised the manuscript to improve clarity and self-containment. Regarding the novel contributions to the PFMLP architecture, we have provided a more detailed explanation in the updated manuscript. The original PFMLP architecture was designed for single-image processing. Our key contribution lies in extending its functionality to handle the Multi-Image Fusion (MFF) task by proposing a new fusion module. Furthermore, we conducted extensive experiments to systematically explore the impact of different parameter choices on model performance, which allowed us to determine the optimal configuration that achieves the best balance between quality and efficiency. We believe these revisions will better highlight our specific contributions and provide a clearer understanding of how we innovated upon the existing architecture.
Comments 3: Similar to the second point, when introducing its core architecture (e.g., the PFMLP network), the paper directs readers to an external reference [27] for a "comprehensive description." While common in academic writing, to improve the paper's readability and self-contained nature, I recommend providing more key details within the manuscript itself.
Response 3: Please refer to the response for the previous question.
Comments 4: The loss function mentioned in the paper is composed of L1 loss and SSIM loss. However, regarding the selection of the weights (alpha and beta) for these two loss components, the authors only mention "extensive empirical evaluation." The lack of an in-depth analysis of the impact of different weight combinations may raise questions about the stability and optimality of the model's training.
Response 4: Thanks for your valuable feedback. We appreciate your insightful comment on the loss function and the selection of weights. We agree that a thorough analysis of the weights (alpha and beta) would provide more insight. In our work, the choice of using a combination of L1 loss and SSIM loss with specific weights was guided by the findings of the paper "Loss functions for image restoration with neural networks" by Zhao et al. (2016). This specific combination has also been successfully adopted in other notable works, such as SwinMFF, demonstrating its effectiveness and stability for image restoration tasks. During our experiments, we also found this particular loss function combination to be effective and stable for the problem we're addressing. While we understand the benefit of a deeper analysis, a detailed exploration of the loss function weight ratios is not a central part of our current research focus. Therefore, to maintain the paper's primary focus on our core contributions, we hope you can understand and allow us to keep the current content as is.
Reviewer 4 Report
Comments and Suggestions for Authors
See "Comments.pdf"

Round 2
Reviewer 4 Report
Comments and Suggestions for Authors
The authors have addressed the reviewers' comments and revised the manuscript accordingly. However, several minor issues remain in the manuscript, such as incomplete reference citations.
Therefore, it is suggested that the authors conduct a thorough review to ensure all references are complete and properly formatted.
Author Response
Comments 1: The authors have addressed the reviewers' comments and revised the manuscript accordingly. However, several minor issues remain in the manuscript, such as incomplete reference citations.
Therefore, it is suggested that the authors conduct a thorough review to ensure all references are complete and properly formatted.
Response: Thank you for your careful review and constructive feedback. We appreciate your suggestion to conduct a thorough review of the manuscript.
Following your recommendation, we have performed a comprehensive review of the entire manuscript and have made several revisions. For example, we have removed the URL from the abstract and, where a URL is referenced in the text, we have added the date of our last successful access. We have also addressed other minor issues throughout the manuscript.
Regarding the incomplete and improperly formatted references you noted, we believe this may be due to differences in LaTeX compilation. We plan to address these formatting issues during the proofreading stage, in collaboration with the editorial team. We hope this approach is acceptable.
We are confident that the revised manuscript is much improved, and we appreciate your guidance in this matter. We look forward to any further comments you may have.